behaviour, evolution, cognition

social learning, mate copying, personality, mosquitofish, *Gambusia holbrooki*

**Author for correspondence:**
Sabine Nöbel
e-mail: sabine.noebel@iast.fr

# The importance of population heterogeneities in detecting social learning as the foundation of animal cultural transmission

Sabine Nöbel[1,2], Xiaobo Wang[2], Laurine Talvard[2], Juliette Tariel[2], Maëva Lille[2], Julien Cucherousset[2], Myriam Roussigné[3] and Etienne Danchin[2]

[1]Université Toulouse 1 Capitole and Institute for Advanced Study in Toulouse (IAST), Toulouse, France
[2]Laboratoire Évolution et Diversité Biologique (EDB UMR 5174), Université de Toulouse Midi-Pyrénées, CNRS, IRD, UPS, 118 route de Narbonne, 31062 Toulouse, France
[3]Centre de Biologie Integrative (CBI), Centre de Biologie du Développement (CBD), UMR5547, CNRS, Université de Toulouse III Paul Sabatier, 118 route de Narbonne, 31062 Toulouse, France

SN, 0000-0002-1850-8895; JT, 0000-0003-1552-4088; JC, 0000-0003-0533-9479; MR, 0000-0002-4240-4105; ED, 0000-0002-5013-9612

High levels of within-population behavioural variation can have drastic demographic consequences, thus changing the evolutionary fate of populations. A major source of within-population heterogeneity is personality. Nonetheless, it is still relatively rarely accounted for in social learning studies that constitute the most basic process of cultural transmission. Here, we performed in female mosquitofish (*Gambusia holbrooki*) a social learning experiment in the context of mate choice, a situation called mate copying (MC), and for which there is strong evidence that it can lead to the emergence of persistent traditions of preferring a given male phenotype. When accounting for the global tendency of females to prefer larger males but ignoring differences in personality, we detected no evidence for MC. However, when accounting for the bold–shy dichotomy, we found that bold females did not show any evidence for MC, while shy females showed significant amounts of MC. This illustrates how the presence of variation in personality can hamper our capacity to detect MC. We conclude that MC may be more widespread than we thought because many studies ignored the presence of within-population heterogeneities.

## 1. Introduction

Many empirical studies have documented a high level of heterogeneity in wild populations, which often are composed of individuals that strongly differ in various ways, male versus females, dispersers versus residents [1], social versus individual learners, bold versus shy [2,3], etc. In agreement with this finding, many theoretical studies adopting an evolutionary stable strategy approach by pitting two or more strategies against each other report instances when more than one strategy persist in the system, often in a form of a density dependence process (e.g. [4]). It is thus highly likely that all populations, be they natural or in the laboratory, are heterogeneous in that they encompass several contrasting strategies.

Such heterogeneities can have drastic consequences in terms of population dynamics and thus can change the adaptive and evolutionary fate of populations. In the context of mate choice, if populations had only one type of strategy, for instance, 'choose your mate based on private information', this would imply high sampling costs before choosing a mate. Besides high energetic costs, this sampling strategy includes predation risk, the risk of harassment, as well as a

high risk of failure due to sampling errors (reviewed in [5]). Using social information to assess the quality of mates can increase accuracy and reduce the costs of gathering private information on a number of potential mates [6]. However, to receive valid social information, the observing individual needs to rely on honest signalling from the sender and slow environmental changes as otherwise the information is quickly outdated. Furthermore, it may create competition over resources if many individuals rely on the same information. Since the use of both private and social information has advantages and disadvantages for the individual as well as for the group, which depend, among other things, on how many individuals in a population use each strategy, both strategies are expected in natural populations.

An individual's tendency to use one source of information over others depends on intrinsic characteristics that can vary among individuals [7,8]. A major source of within-population heterogeneity is personality that involves trait variation that is highly consistent across time and/or context within individuals [9,10]. In particular, variation in personality can correlate with variation in social information use. One of the most studied personality traits is boldness, which is a major axis of behavioural variation [3,11–15]. Typically, bold individuals are active in a novel environment (presumably exploring it) and are more likely to take risks rather than retreating or freezing [2]. By contrast, shy individuals respond to unfamiliar situations by fleeing, retreating, becoming cautious, quiet or inactive [15] and observing. In doing so, shy individuals have the opportunity to gather information. More generally, behavioural differences along the bold–shy axis may profoundly correlate with courtship, foraging, feeding and adaptability to environmental change, as well as to the capacity to extract information from the environment [16–19].

With few, but significant, exceptions (e.g. [8,20–22]), the question of the impact of population heterogeneity has been neglected often in the context of social learning that constitutes the fundamental process of cultural transmission [23–25]. Most studies of social learning implicitly assume that the study population is homogeneous and composed of social learners. However, personality and particularly the bold–shy gradient that seems to be universal in vertebrates [2], as well as at least in some invertebrates [26], constitutes a potentially important type of heterogeneity when studying various forms of social learning such as mate copying (MC) (also called mate-choice copying). MC corresponds to situations when the observation of sexual interaction involving potential partners influences the future mate choice of observer individuals [6,24,27].

Typically, MC occurs when an observer individual (usually a female) alters its mating preference in favour of mates they previously observed being chosen by conspecifics [28–30]. It has been experimentally demonstrated in a suite of birds (e.g. [31–34]), mammals (e.g. [35–37]) and fish (reviewed in [5]), as well as in *Drosophila melanogaster* (e.g. [25,38,39]). In fish, most evidence for MC comes from species of the genus *Poecilia*, which are easy to maintain in the laboratory, including the sailfin molly *Poecilia latipinna* [40–42] and the guppy *Poecilia reticulata* [29,43–45]. Despite the potential of personality to affect information gathering by members of a population and although an effect of sociability on MC has been described previously [22,23], the impact of such individual heterogeneities has rarely been accounted for in the study of MC.

Here, we perform a mate-copying experiment in female mosquitofish (*Gambusia holbrooki*), while accounting for the personality of observer females. Our goal was to study the potential impact of population heterogeneity on MC that constitutes the most basic mechanism for the emergence of cultural traditions in mating preferences [25,46–48]. We also account for potential confounding effects such as the extent of size difference between stimulus males. We hypothesized that MC would be influenced by both the personality of the observer female and the amplitude of the relative size difference between the stimulus males. More specifically, we predicted shy individuals to behave as information gatherers and bold individuals as kind of 'blind' explorers, i.e. that MC would be stronger in shy than bold individuals.

## 2. Methods

### (a) Study species
The eastern mosquitofish (*G. holbrooki*) belongs to the family Poeciliidae where sexes do not differ in colour but show sexual size dimorphism. Males can also differ in body length: males born early in the year are smaller than males born in the previous year [49]. Females prefer larger males [50–52] possibly because larger males harass them less [52,53]. *Gambusia* is often found in mixed-sex groups or in schools [50], which give them opportunities for social learning.

### (b) Fish maintenance
We used mature mosquitofish caught by hand netting in summer 2015 and 2016 in Lake Lamartine, Roque-sur-Garonne, France (43°30′31.3″ N 1°20′18.1″ E) [54]. Lake Lamartine is part of a large network of gravel pit lakes in the floodplain of the Garonne River with a fish assemblage composed of native and non-native fish species [55]. Fish were housed in mixed-sex tanks (60 cm × 40 cm × 30 cm) with a constant temperature of 24°C and a 14 : 10 h light : dark cycle. They were fed twice a day *ad libitum* with flake food (Novobel JBL) or frozen *Daphnia* (Midisel). About 7 days before experiments, fish were sexed and kept in same-sex groups under the same conditions. Fish were returned to the stock tanks after the experiment. None of the methods used for our experiments involved regulated procedures.

### (c) Mate-copying experiments
The experimental set-up was adapted from previous fish studies [40,41]: a large test tank (50 cm × 30 cm × 30 cm) and four small stimulus tanks (15 cm × 10 cm × 25 cm) standing two by two at each smaller end of the large tank (figure 1). A mate-choice zone (15 cm × 15 cm) was marked in front of the stimulus tanks. The water in the tanks was 20 cm deep and had a constant temperature of 24°C. The backsides of the tanks were covered with white plastic to avoid any disturbances from outside.

### (d) Experimental design
The design encompassed a series of steps in which the observer female remained in the large test tank (figure 1). First, a small male and a large male were placed in two of the small tanks, diagonally from each other to maximize distance. Opaque screens (white plastic boards) were inserted between the test tank and the four small tanks to prevent observer females from seeing stimulus males. All three fish could acclimatize for 20 min. Second, the observer female was gently placed in a clear glass square tube (10 cm × 10 cm × 35 cm) in the middle of

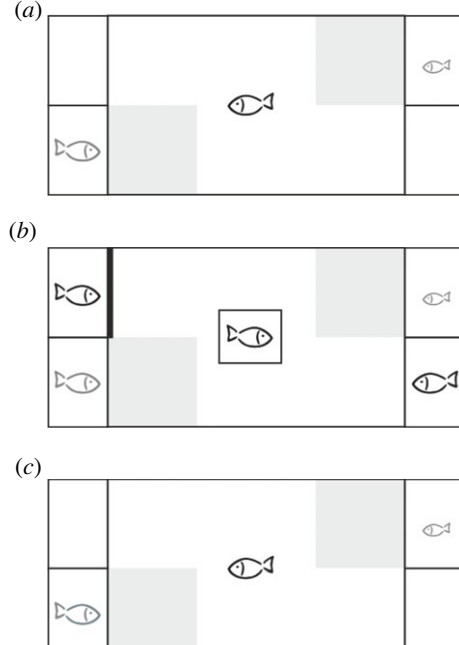

**Figure 1.** Top view on the experimental set-up and design of the mate-copying experiment. (*a*) First mate-choice test: the grey fields mark the two mate-choice zones. The observer female (black) is in the large tank and two males (grey), a large (lower left) and a small (upper right), are placed diagonally in one of the small stimulus tanks at each end of the large tank. (*b*) Demonstration phase for 10 min. A stimulus female (black) is placed in a separate tank next to the small male (lower right tank) so that it is visible to the observer female. A pseudo-stimulus female (black) is also placed in a separate tank next to the large male (upper left) but behind a screen (thick black bar) and thus not visible to the observer female. (*c*) Second mate-choice test (like the first test).

the large tank and the opaque screens were removed simultaneously. The observer female was allowed to watch the small and large stimulus males for 10 min. Third, the central glass square was removed freeing the observer female and the time the female spent within the mate-choice zones in front of each stimulus male was recorded with stopwatches for 10 min (first part of the first mate-choice test; figure 1*a*). Fourth, the opaque screens were inserted, and the observer female was placed back into the clear glass square in the middle of the large tank. The small tanks containing the stimulus males were then swapped between the two corners to control for potential observer female's side-biases (SBs). The opaque screens were then removed, and the observer female had 5 min to observe the stimulus males before it was released from the glass square for a second mate-choice test of 10 min.

The times spent by the observer female in front of a given stimulus male in the two mate-choice tests were added for each stimulus male separately. The times the observer female spent within the mate-choice zones of a given male was used as a score of male attractiveness: the observer female was considered to prefer the stimulus male closest to which she spent more time during the 20 min of the total mate-choice test. Relative proximity to the males has been shown to correlate positively with the mating probability [56–59] and, thus, constitutes a good proxy of preference.

Trials in which observer females spent more than 90% of the time in total in one of the mate-choice zones were considered as side-biased and were stopped at this stage as is common in mate-copying experiments [60,61]. Fifth, we inserted opaque screens and the observer female was placed back into the glass square in the middle of the large tank. One stimulus female was placed next to each male, but only the one close to the small

male was visible to the observer female; the female next to the large male was hidden to the observer by an opaque plastic screen (thick line in figure 1*b*). To minimize disturbance, all female insertions were performed in the presence of another screen separating the small tanks from the central tank. The observer female then could see a stimulus female near the small male and the large male apparently alone for a 10 min demonstration (figure 1*b*). At the end of demonstration phase, the screens were inserted again, and the stimulus females were removed. Sixth, we removed the opaque screens, released the observer female from the glass square and started the second mate-choice test which duplicates the first one in all aspects (figure 1*c*). Seventh, the observer female was moved to a maze tank to assess its personality (see 'Boldness test' section). Finally, male and female body lengths were measured from the tip of the snout to the caudal peduncle to the nearest mm.

The average body length of observer females in the mate-copying experiment was 30.2 mm ± 0.4 mm. In the mate-copying experiment, stimulus and pseudo-stimulus females were matched for body length (Kruskal–Wallis test: $X = 19.996$, d.f. = 28, $p = 0.865$) with an average body length of 31.1 mm ± 0.4 mm and 30.9 mm ± 0.4 mm, respectively. Male sizes ranged from 16 mm to 37 mm. The relative size differences between the two males used for a given test varied from 0% to 31%. Further details about the time spent by observer females in front of the stimulus males are provided in the electronic supplementary material, tables S1–S3.

### (e) Control 1: consistency in mate choice in the absence of social information during demonstrations (C1)

In control 1, observer females did not receive any social information during the demonstrations. It followed the same protocol except that during demonstrations both stimulus females were hidden to the observer female by opaque plastic screens placed between the small tanks and the central one, and thus not visible to the observer female.

In this control, the average body length of observer females was 31.9 mm ± 0.7 mm. Both pseudo-stimulus females were matched for body length (Kruskal–Wallis test: $X = 14.239$, d.f. = 24, $p = 0.941$) with an average body length of 32 mm ± 0.7 mm and 32.1 mm ± 0.7 mm, respectively. Male sizes reached from 17 mm up to 40 mm, with size differences between the two males of a given test varying from 0% to 33%.

### (f) Control 2: group size effects (C2)

This control tested whether females tend to associate with the males that they saw with a second conspecific (group size of 2) versus apparently alone (group size of 1) during the demonstration. Control 2 duplicated the mate-copying experiments in all points except that all males were replaced by females.

In this control, the average body length of observer females was 32.9 mm ± 0.4 mm. Stimulus females and pseudo-stimulus females were matched for body length (Kruskal–Wallis test: $X = 8.8752$, d.f. = 18, $p = 0.963$) with an average body length of 33.4 mm ± 0.3 mm and 33.4 mm ± 0.3 mm, respectively. Stimulus female sizes reached from 26 mm up to 45 mm, with size differences between the two females of a given test varying from 0% to 32.6%.

### (g) Social learning index

For each observer female, we first calculated scores separately for the first (MCT1) and the second (MCT2) mate-choice test using the following formula: MCT = tS/(tS + tL), where tS is the time spent in front of the smaller male and tL the time spent in front of the larger male. Then, we calculated for each observer

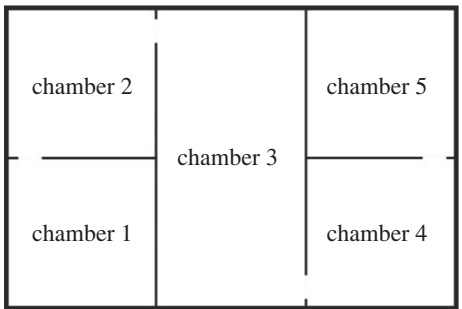

**Figure 2.** Top view on the experimental test maze to determine boldness as exploratory behaviour. The tank was separated into five chambers by four dark grey plastic boards, each with an opening in their middle for females to explore all chambers. The female was gently placed into chamber 1 at the beginning of the test and given a maximum of 10 min to reach chamber 5.

female a social learning score (SLS) as the difference in scores of the first and second mate-choice test (MCT2 − MCT1). Negative values indicate a decrease in time spent in front of the small males (no MC) and positive values indicate an increase in time spent in front of small males (MC). The mean of the SLS of all females within the same condition was called social learning index (SLI).

### (h) Observer female side-bias

Observer females whose total time spent on the mate-choice zone of one side represented more than 90% of the total time spent in the two mate-choice zones during the first mate-choice test (independently from the fact that we swapped the males in the middle of the mate-choice test) were removed from our study because they showed a strong SB. Nonetheless, for all other observer females, we calculated a SB for each female as SB = |(L1 + L2)/ (L1 + L2 + R1 + R2)| − 0.5, where L1 and L2 are the time spent in the left mate-choice zone and R1 and R2 are the time spent in the right mate-choice zone respectively in the first (L1, R1) and second part (R2, L2) of the first mate-choice test. We subtracted 0.5 from the absolute value of the second term of this equation in order to be able to add the left and right deviations from the expected random distribution of 50% of their time in each mate-choice zone.

### (i) Boldness test

We measured boldness as exploratory behaviour in a test maze immediately after each MC or control experiment. Tests were performed in a tank (30 cm × 20 cm × 20 cm) separated in five chambers (chamber 3: 10 cm × 20 cm; the other four chambers: 10 cm × 10 cm) by four dark grey plastic boards (figure 2). Each plastic board had an opening (3 cm in diameter) allowing fish to move across the maze from chamber to chamber. The sides of the test tank were covered with opaque foil to avoid any disturbances from outside. The water in the tanks was 10 cm deep and had a constant temperature of 24°C.

First, a female was gently placed into chamber 1. We then classified individuals that reached the last chamber within 10 min as bold and individuals that did not as shy in a binary fashion. Due to this dichotomy, we had 30.5% (87) bold and 69.5% (198) shy females (for more details, see the electronic supplementary material, tables S1 to S3).

### (j) Statistical analyses

Details about the time spent by observer females in front of the stimulus males are shown in the electronic supplementary material, tables S1–S3. Data analyses were carried out with the

R software (v. 3.3.3) [62]. In total, we tested 343 females but excluded 58 females from the analyses because they showed a SB of more than 90% in the first mate-choice test. Thus, we ended up with 140 females in the mate-copying treatment, 71 in the control for consistency in mate choice and 74 in the control for shoaling behaviour (for more details, see the electronic supplementary material, tables S1 to S3). We used a linear regression model (LM) with a quadratic term. We started with a univariate analysis testing the impact of the experimenter and body length of the observer female as potential confounding effects. As neither the experimenter (E.D., J.T., L.T., M.L., S.N., X.W.; LM: $F = 0.912$, $p = 0.474$) nor the body length of the observer female (LM: $F = 0.251$, $p = 0.617$) significantly affected SLSs, we did not include them in the main model. The starting model included the SLS as dependent variable (Shapiro–Wilk test: $W = 0.995$, $p = 0.481$) and treatment (MC, control 1 (C1), control 2 (C2)), personality (shy versus bold), male size-ratio (%; arcsin-square-root-transformed) plus its square (as we expected a nonlinear relationship) and the SB as fixed effects. The starting model also included all possible interactions between treatment, personality and male size-ratio. The significance of fixed effects was tested using Wald chi-square tests implemented in the ANOVA function of the car package [63]. We applied a stepwise backward selection method using $p$-values, by dropping out non-significant effects one by one, starting with the highest order interactions. We used the Akaike information criteria (AIC) [64] to determine the final model.

## 3. Results

To study the potential impact of population heterogeneity on MC, we performed a mate-copying experiment in female mosquitofish, while accounting for the bold–shy personality of observer females. We also addressed potential confounding effects such as the extent of size difference between stimulus males. Without any social information, females significantly preferred larger males in the first mate-choice test of the mate-copying and control 1 treatment (i.e. prior to receiving any social information; Wilcoxon test: $N = 184$, $V = 12936$, $p < 0.0001$; for more information, see electronic supplementary material, table S4). To test for MC, the starting statistical model used the SLSs as the dependent variable and included all potential independent effects as described above. The interaction between treatment, personality and male size-ratio was close to significant (LM: $F = 2.930$, $p = 0.055$) and had to be kept in the model because removing it increased the AIC (Delta AIC 2.11). As we were mainly interested in testing the significance of our treatment effect, and as personality (shy, bold) was involved in the third-order interaction, we thus continued statistical analysis separately for bold and shy observer females.

In bold observer females ($n = 87$), the interaction between treatment and size difference between males was non-significant (LM: $F = 2.121$, $p = 0.127$). After removing that interaction, none of the main effects were significant (SB, LM: $F = 0.175$, $p = 0.677$; treatment, LM: $F = 1.391$, $p = 0.255$; male size-ratio, LM: $F = 2.116$, $p = 0.150$; male size-ratio², LM: $F = 1.684$, $p = 0.198$), providing no evidence for MC in bold observer females.

By contrast, in shy observer females ($n = 256$), the interaction between treatment and male size-ratio was close to significant (LM: $F = 2.838$, $p = 0.061$) and its removal slightly increased the AIC (Delta AIC 1.58). As treatment was involved in the second-order interaction of shy females, to

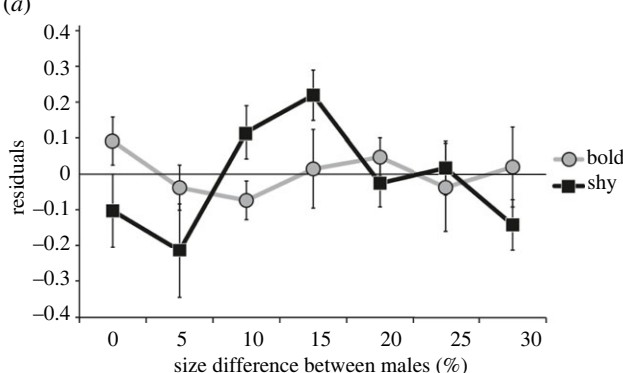

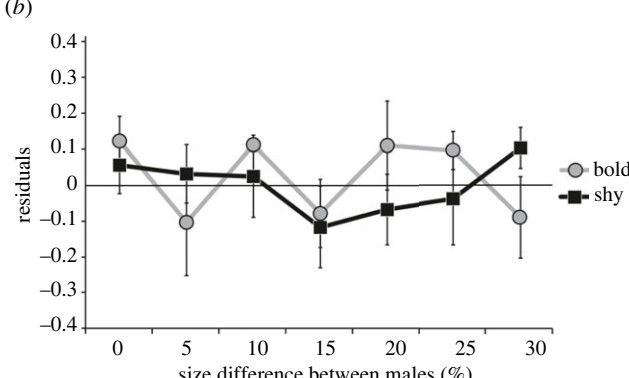

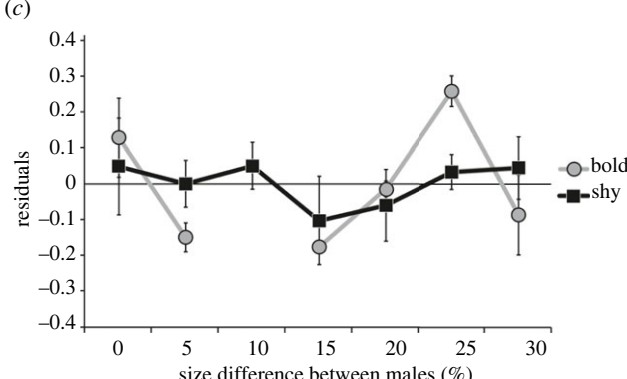

**Figure 3.** Residual plots for (*a*) mate-copying trials, (*b*) control 1 for consistency in mate choice trials in the absence of social information, and (*c*) control 2 for group size effects trials. Null or negative residuals indicate no copying while positive values indicate copying. Numbers next to the points indicate the sample size for each specific size-ratio category.

better analyse the meaning of this interaction, we then split this sub-dataset in three parts, one with control 1 trials, one with control 2 trials and one with mate-copying trials. For each of these, the starting models included the three remaining effects: male size-ratio, male size-ratio² and the SB. For control 1, we found no significant effect(s) explaining the SLS (male size-ratio, LM: $F = 0.851$, $p = 0.362$; male size-ratio², LM: $F = 0.091$, $p = 0.765$; SB, LM: $F = 0.371$, $p = 0.546$; figure 3*c*). Similarly in control 2, none of the tested effects was significantly related to the SLSs (SB, LM: $F = 0.468$, $p = 0.545$; male size-ratio, LM: $F = 0.541$, $p = 0.380$; square male size-ratio, LM: $F = 0.620$, $p = 0.249$; figure 3*b*). Thus, in both controls, shy observer females' choice was consistent between the first and second mate-choice tests, which revealed that their choice did not change significantly between these two tests. Contrastingly, all three variables were significant for

the mate-copying trials (male size-ratio LM: $F = 6.086$, $p = 0.016$; male size-ratio² LM: $F = 4.590$, $p = 0.035$; SB LM: $F = 8.386$, $p = 0.005$; figures 3 and 4).

Therefore, shy observer females of the mate-copying treatment group showed a significant tendency to perform MC, and this behaviour was affected by male size-ratio and SB (figure 4). MC was not detected when the size difference between the two stimulus males was too large (greater than 20%) or too subtle (less than 10%) but was found at intermediate male size-ratios (between 10% and 20%) and the level of MC significantly increased with observer females' SB (figure 4). Thus, the highest SLSs occurred at intermediate male size-ratios.

Finally, to test the treatment effect *per se*, we performed a final analysis in shy females to test for the treatment effect within the trials of intermediate male size-ratio (10–20%). The starting model of the SLSs included the treatment effect (MC versus controls), male size-ratio and its square (to model a potentially nonlinear relationship) and SB, plus the interaction between treatment and male size-ratio. The interaction between treatment and male size-ratio was non-significant (LM: $F = 0.112$, $p = 0.739$). As expected, the SB, male size-ratio plus its square were non-significant as this analysis did not incorporate the low and high male size-ratios (SB, LM: $F = 1.663$, $p = 0.2$; male size-ratio, LM: $F = 0.097$, $p = 0.765$; male size-ratio², LM: $F = 0.045$, $p = 0.083$). However, the treatment effect was significant (LM: $F = 7.692$, $p = 0.007$). Thus, shy females' behaviour differed significantly between the mate-copying treatment and the control treatments when using trials with intermediate male size-ratios.

## 4. Discussion

The aim of our study was to test whether MC exists in *G. holbrooki* females, while accounting for potential confounding effects. In particular, we were interested in the potential effect of personality as they both had the potential to affect MC. As expected, we found no evidence for MC in bold females but found evidence for MC in shy females. In shy observer females, the role of male size-ratio appeared to be quadratic as both the male size-ratio (positive relationship) and its square (negative relationship) were significant. Accordingly, the relationship of SLSs with male size-ratio was bell shaped suggesting that shy observer females copied the choice of other females mainly when the size difference between stimulus males was intermediate (figure 3*a*). Hence, MC in shy females was significant when the male size-ratio was between 10% and 20% with a maximum around 15% (figure 4). Below male size-ratio of 10% or above 20%, shy females did not seem to copy. Finally, the SB of the observer female calculated during the first mate-choice test appeared to be positively related to MC in shy females indicating that highly side-biased shy females were the most likely to copy.

Concerning the effect of personality, we expected shyer females to be more prone to copying than bold ones, which we found. Our prediction resulted from the observation that shy individuals behave as if gathering information before action, while bold individuals act immediately. This result is consistent with recent findings in zebra finches where females that are more active in a new environment

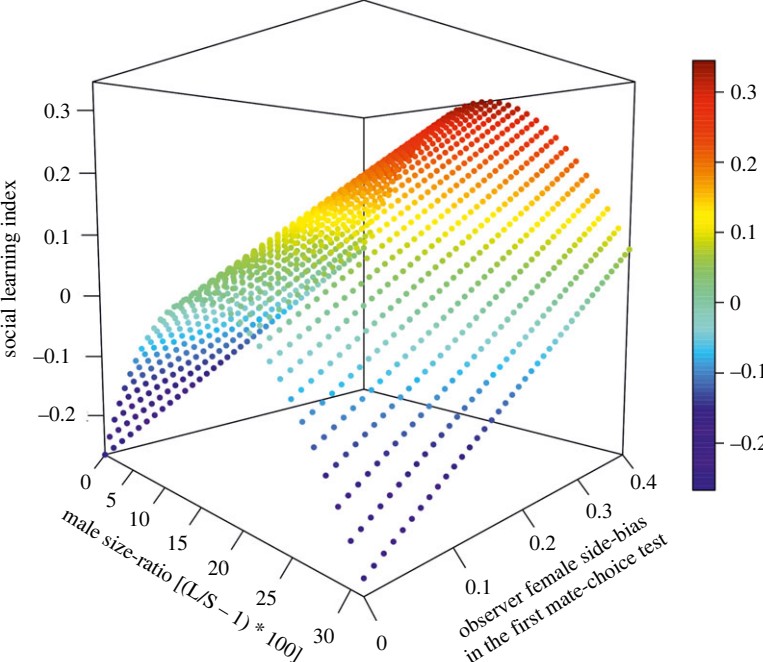

**Figure 4.** Predicted SLSs in shy females of the mate-copying treatment according to our selected statistical model, including the effects of male size-ratio (bottom left axis), its square and the SB of the observer female (bottom right axis). Here, the SLI is social learning values predicted by the model accounting for these effects. Those predicted values are bell shaped as revealed by the quadratic term, and the level of the surface globally goes up as the SB during the first preference test (see Methods). Note that here we use the SB during the first mate-choice test in shy individuals of the mate-copying treatment, but that using the SB during the second mate-choice test gave similar results. (Online version in colour.)

were less likely to copy in mating and foraging situations [8]. A benefit of MC is that it allows observer females to avoid sexual harassment. By assessing the attractiveness of potential mates through their mating success with other females, females can thus gain information about males and avoid aggressive males [61]. As sexual harassment in mosquitofish can be quite high, females might really benefit from MC in that species. This also matches with the finding of White *et al.* [23] that sociability in guppy females predicts mate-copying tendency. They defined sociability as proclivity to be with other females, which is at the same time a protection from male harassment. When Gomes-Silva *et al.* studied MC in *Gambusia affinis* males (a species in which males are harassing females), they did not find a correlation between personality measured as boldness, activity and shoaling tendency with MC [22].

Concerning the effect of the male size-ratio that we manipulated during the experiments, we expected MC to diminish when size differences between the two males became too high, which we found. Perhaps in such circumstances, the social information was not strong enough to override the innate preference for large males. In a study in the sailfin molly, demonstration length had to be increased from 10 to 20 min for females to copy the choice of a small or heterospecific male [40,60]. In agreement with this study, our 20 min demonstrations proved to be long enough to override the general preference for larger males up to a male size-ratio of around 20%, but not beyond. This result is similar to the one obtained in guppy females, *P. reticulata* [65], where females copy the choice of stimulus females when males differ in the amount of orange by 24% or less but did not copy when this difference reached 40%. Due to the available male sizes, we could not go beyond male size-ratios of 30%, but our results are consistent with those of the guppy in

that MC could not override the innate preference for large males when male size difference was too large. Concerning low male size-ratios, we had different predictions according to whether observer females can identify very similar looking males individually. If they did so, we expected MC to persist when the size difference between the males became very small to inexistent, as is the case in the related sailfin molly [41,66]. Also several theoretical models show that females should be more likely to copy if males are very similar (e.g. [28]) because the social environment provides additional information about the attractiveness and success of males to females that are uncertain about their decision. On the other hand, if observer females cannot tell apart two males of very similar sizes, then we expected MC to diminish with increasing similarity in male size. Our results are in agreement with that second prediction, suggesting that *Gambusia* females may not be able to tell two very similar males apart.

Concerning the SB, we had no prediction about its potential role beyond the fact that, as in most previous studies (e.g. [60,61]), we stopped trials in which the observer female spent more than 90% of the total time she spent in mate-choice zones in one mate-choice zone during the first mate-choice test. It was generally assumed that these females were not currently interested in males or were frightened. Nonetheless, we can speculate *a posteriori* about the biological significance of this parameter. As stated above, we could expect a positive correlation between shyness and SB if the latter simply reveals shyness. However, in our dataset, the SB of shy and bold females did not differ statistically (Mann–Whitney U test: $W = 2200$, $p = 0.886$). Furthermore, we found that both personality and the SB simultaneously correlated with MC (figure 4). This suggests that these two parameters describe independent components of the

phenotype as supported by the fact that removing one of them from the analysis did not improve the model. Our *a priori* hypothesis was that highly side-biased females would gather no social information on males either because they were less motivated or because they were too scared to observe. This is why, as in many other studies, females with side preferences over 90% were excluded. Also, a high SB might result from the absence of preference for any of the males. Females without an initial preference for one of the males could be more likely to use social information and are probably better social learners, and thus tend to copy more efficiently as our data suggest (figure 4). Therefore, not only shy individuals with high SBs may not be frightened, but they seem to be good information gatherers and would support the 'copy-when-uncertain' hypothesis. However, note that all these interpretations were formulated *a posteriori* in order to interpret an unexpected result. This SB may reflect an overlooked component of personality and we suggest that future fish studies should account for this parameter.

Finally, the finding that highly side-biased shy females tested with intermediate male size-ratio strongly differed from the two controls under the same conditions supports the interpretation that it is the social information provided during the demonstration of the mate-copying treatment that fostered the increase of time spent in the small male mate-choice zone between the first and the second mate-choice test.

We note that we are aware of the possibility that the regression toward the mean (RTM) fallacy first mentioned by Galton [67] might affect this type of MC study as RTM occurs in all kinds of studies with repeated measures of the same individual [68]. This effect is stronger if individuals are categorized based on their first performance. Because of this, we allocated the observer females randomly to the treatment and control groups and decided *a priori* that we would always show the demonstrator female next to the smaller male independently of what happened in the first mate-choice test. Thus, the RTM should affect all groups equally, thus allowing proper comparison [69].

Furthermore, we had designed our experiments from the very beginning in order to incorporate the effects of personality, male size-ratio and SB into our study, as we speculated that they might play an important role. Interestingly enough, we also found that if we had ignored the three potential confounding effects of personality, male size-ratio and female SB, we would have concluded that there is no MC in that species and probably would have been unable to publish that negative result. This observation suggests that in fact MC is likely to be more common than usually envisioned because such studies usually do not control for the effect of potential confounding effects. If personality-dependent differences in information use proved to be valid, this would further imply that the incidence of MC might be much broader than usually thought. This would also explain why not all individuals in a population copy and some individuals seem to mainly rely on private information. Theoretical models of copying in mate choice assume strong positive frequency-dependent selection, which quickly leads to uniform mating decisions and possibly extinguishes certain phenotypes. However, usually we see a broad variety of phenotypes in nature, although there has been no explanation for a mechanism that maintains

variation in mating preferences. Our study suggests that the use of social information, and thus preferences for particular mates, may be influenced by personality that may result from selection in another context. Copying by shy individuals could be a source of positive frequency-dependent selection at the population level, but the aversion of bold females to copying prevents the population from committing to a single phenotype, thus maintaining an independent source of mating preferences.

Our study highlights that *G. holbrooki* females can use social information to choose mates and shows that personality and experimental parameters like male size-ratio can influence MC. Although the importance of MC would need to be evaluated further in the field, the reported mosquitofish mate-copying abilities set the stage for a role of social processes in the evolutionary trajectories of populations in this species. As MC constitutes a fundamental mechanism of cultural transmission and inheritance, this would in turn suggest that the existence of cultural processes might be far more common than envisioned. After all, a recent study using the MC paradigm concluded the potential existence of a cultural transmission of mating preferences even in the minute *Drosophila melanogaster* [25]. Our point here is that the study of MC has much broader ecological and evolutionary consequences than the mere beauty of a punctual learning process; it may reveal the existence of a general mechanism of non-genetic inheritance with all its range of potential consequences for evolution.

**Ethics.** All animals were handled in accordance with the guidelines from the European directive on the protection of animals used for scientific purposes (2010/63/UE), French Décret 2013-118. Fish were caught from Lake Lamartine (Arrêté Prefectoral 27 January 2014, 6 July 2016) and M.R. has received an authorization to experiment on vertebrate models (no. 311255556) from the 'Direction Départementale de la Protection des Populations de la Haute-Garonne'. All efforts were made to minimize the number of animals used and their suffering, according to the guiding principles from the Décret 2013-118.

**Data accessibility.** Data supporting this paper are available from the Dryad Digital Repository: https://doi.org/10.5061/dryad. g79cnp5s0 [70].

The data are provided in the electronic supplementary material [71].

**Authors' contributions.** S.N.: conceptualization, data curation, formal analysis, funding acquisition, investigation, supervision, visualization and writing—original draft; X.W.: funding acquisition, investigation and writing—review and editing; L.T.: investigation and writing—review and editing; J.T.: investigation and writing—review and editing; M.L.: investigation and writing—review and editing; J.C.: resources and writing—review and editing; M.R.: supervision and writing—review and editing; E.D.: conceptualization, formal analysis, funding acquisition, investigation, resources, supervision and writing—review and editing.

All authors gave final approval for publication and agreed to be held accountable for the work performed therein.

**Conflict of interest declaration.** We declare we have no competing interests.

**Funding.** This work was supported by the 'Laboratoires d'Excellences (LABEX)' TULIP (ANR-10-LABX-41), as well as ANR funded Toulouse Initiative of Excellence 'IDEX UNITI' (ANR11-IDEX-0002-02). Work of E.D. and S.N. was also supported by the Soc-H2 ANR project (ANR-13-BSV7-0007-01 to E.D.). S.N. acknowledges IAST funding from the French National Research Agency (ANR) under the Investments for the Future (Investissements d'Avenir) programme (ANR-17-EUR-0010) as well as a Marie Curie PRESTIGE grant (PRESTIGE-2014-1-0005). X.W. was supported by a scholarship from the China Scholarship Council (file no. 201206620006).

**Acknowledgements.** We thank Remy Lassus for his help with fish sampling.

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
