## [Peer Review File · Proceedings of the Royal Society B: Biological Sciences]

Review History

RSPB-2021-1567.R0 (Original submission)

Review form: Reviewer 1

Recommendation

Reject – article is not of sufficient interest (we will consider a transfer to another journal)

Scientific importance: Is the manuscript an original and important contribution to its field?

Acceptable

General interest: Is the paper of sufficient general interest?

Good

Quality of the paper: Is the overall quality of the paper suitable?

Good

Is the length of the paper justified?

Yes

Should the paper be seen by a specialist statistical reviewer?

No

Do you have any concerns about statistical analyses in this paper? If so, please specify them explicitly in your report.

Yes

It is a condition of publication that authors make their supporting data, code and materials available - either as supplementary material or hosted in an external repository. Please rate, if applicable, the supporting data on the following criteria.

Is it accessible?

Yes

Is it clear?

Yes

Is it adequate?

Yes

Do you have any ethical concerns with this paper?

No

Comments to the Author

In the manuscript entitled „The importance of population heterogeneities in detecting social learning as the foundation of animal cultural transmission” the authors report about a study in which the degree of female mate-choice copying was linked to the copying female’s personality type. The authors found that only distinct personality types copy the mate choice of other females in their experimental design, whereas other personality types were not responsive to this specific type of social learning. An interesting aspect that the authors stress is that behavioural differences among personality types can potentially blur the results in behavioural studies, which can lead to misinterpretations if this ubiquitous heterogeneity is ignored.

The manuscript is well-written and was easy to follow.

My advice for improvement of the manuscript is listed in a point-by-point fashion below:

The statement of novelty of the study seems very exaggerated given that a bunch of studies exist that investigated the link between socially-dependent mate choice (also in particular mate choice copying) and personality before. Those studies are not even cited although they are highly relevant. For example:

Gomes-Silva et al. 2017 “ Does individual variation in male mate choice copying reflect differences in social responsiveness?”

White et al. 2017 “Socialbility affects the intensity of mate-choice copying in female guppies, *Poecilia reticulata*”

Bierbach et al. 2015 “Personality affects mate choice: bolder males show stronger audience effects under high competition...etc.

Since the authors are competent in the study of animal behaviour and, especially, animal personality, I find it rather surprising that they seem to have overseen some of the most relevant studies in their context, which is a major problem that needs to be addressed. I strongly recommend the authors to include the existing literature and evaluate again what aspects of novelty their study provides compared to the already existing ones.

I recommend to replace “mate-copying” by “mate-choice copying”. Even though both terms appear in the scientific literature, the second one is, in my opinion, more correct and less misleading especially for readers unfamiliar with this field of research.

l. 55-56: I understand that the authors want to set the existence of these behavioural heterogeneities in a bigger context, however I felt very distracted by the detailed example about

breeding habitat choice, which is not the topic of this study. I am sure it is possible to find a scenario that is closer to the actual research topic.

l. 118: what does “kept separately” mean? Individually (isolated) or separated by sex but still in groups? Please specify.

l. 212: Why was boldness not measured as a continuous variable? This would come with less information loss.

l. 217: That is a very high proportion of females. Do the authors have any explanation for this? Lightening, room etc.

l. 218: Is the dependent variable normally-distributed? Please show the statistical values.

l. 225: How many variables were included in the starting model? Although the authors used a rather big sample size, too many variables can cause overfitting of the model. The rule of thumb here would be to include variables of the number square root of sample size.

l. 291: Was that really unexpected? If I am not wrong the formula (MCT2-MCT1) that was used to calculate SLS is inherently larger for individuals that do not show much of a preference during the first test. This should be the case for females with strong side-bias. It seems to me that a female, which did have a strong preference in the first mate choice test (before copying), is unlikely to achieve a high SLS given this calculation. Please clarify.

Review form: Reviewer 2

Recommendation

Major revision is needed (please make suggestions in comments)

Scientific importance: Is the manuscript an original and important contribution to its field?

Good

General interest: Is the paper of sufficient general interest?

Good

Quality of the paper: Is the overall quality of the paper suitable?

Good

Is the length of the paper justified?

Yes

Should the paper be seen by a specialist statistical reviewer?

No

Do you have any concerns about statistical analyses in this paper? If so, please specify them explicitly in your report.

No

It is a condition of publication that authors make their supporting data, code and materials available - either as supplementary material or hosted in an external repository. Please rate, if applicable, the supporting data on the following criteria.

Is it accessible?

Yes

Is it clear?

Yes

Is it adequate?

Yes

Do you have any ethical concerns with this paper?

No

Comments to the Author

Review of MS PSPB-2021-1567

This is a very interesting study, that shows how personality can affect the propensity of females to mate copy.

This is important for two reasons: 1) bold-shy dichotomy has mostly been discussed in foraging contexts, here it is shown to be of importance in the propensity of social learning in mate selection context.

2) Theory on the effect of mate choice copying has assumed that this would produce strong positive frequency dependent selection, which would wipe out variation in mating decisions, and there was until now no good reason to assume mechanisms to maintain variation in mate preferences in the population. The pattern shown in this paper adds a source of variation in mate preferences, that has not previously been discussed: copying from shy individuals could yield on a population level a source of positive frequency dependent selection, but the reluctance of bold females to copy prevents it sweep the population, maintaining an independent source of mate preferences.

The paper is overall well written, and I have no structural comments about the study design, or analysis. I do have a couple comments that I think would improve the paper.

1) In the introduction, the authors use a lot of space to draw an analogy to nest-site selection. I find this distracting to the topic at hand, and I would suggest the authors focus on mate selection instead, as I have described above. As it stands now, the authors don't make this point either in the introduction or discussion in the context of mate choice, and I feel this is a lost opportunity on their part.

2) In the results section, a large role is assigned to side bias scores. The authors need to justify this better.

First of all, does the side bias score correlate in any way with the personality score? I would like the authors to look at one or both of the following options: did side bias correlate with the time a female used to reach the last chamber in the boldness test? I am aware that the authors scored the females in a binary fashion as shy or bold, but perhaps they maintained a record of the time score of each female. If not, please check if shy individuals more likely to have a high side bias score?

Secondly, I would like to see some reflection on what a side bias may indicate, more than that an individual spent a lot of time on one side of the tank. In particular I would like the authors to consider that a high side bias score may indicate a lack of choice for a mate, a weak preference. If a female does not have a strong preference for a male, she might play it safe and stay away from males, avoiding predation risks, and/or avoiding male harassment. More importantly for this study, a female with a weak preference may be more likely to copy than a female that showed a strong preference. This point is important, because it relates to the underlying mechanism for why and when individuals use social learning. Shy individuals may not be 'better social learners', but those with a strong side bias are more likely to copy because they are less biased by prior preferences (prior to seeing the demonstrator choices). This also underscores why I would really like to see the author disentangle the effect of the side bias from the shy-bold scores better,

because from this manuscript, I cannot be sure that there is a difference.

3) Lastly, I would like to see figure 3 improved. As it stands, there is no figure in the manuscript that visually shows the strong effect the authors found in the difference between shy-bold individuals and their copying score. I am not blown away by figure 3A, if I were ever to present this mechanism in a lecture or other type of presentation, for instance.

Minor comments:

Ln 62: "Similarly, the addition a small proportion" should read: "Similarly, the addition of a small proportion"

Ln 119: "Fish returned to the stock tanks after experiment." Should read: "Fish returned to the stock tanks after the experiment."

Ln 145: "... prefer the stimulus male close to which she..." should read: "... prefer the stimulus male closest to which she..."

Ln 212: please indicate the N for bold and shy females

Ln 218: please indicate the final N for each experiment after exclusion of the 90% side biased females.

Decision letter (RSPB-2021-1567.R0)

07-Sep-2021

Dear Dr Nöbel:

I am writing to inform you that your manuscript RSPB-2021-1567 entitled "The importance of population heterogeneities in detecting social learning as the foundation of animal cultural transmission" has, in its current form, been rejected for publication in Proceedings B.

This action has been taken on the advice of referees, who have recommended that substantial revisions are necessary. With this in mind we would be happy to consider a resubmission, provided the comments of the referees are fully addressed. However please note that this is not a provisional acceptance.

4) Data - please see our policies on data sharing to ensure that you are complying (<https://royalsociety.org/journals/authors/author-guidelines/#data>).

Sincerely,
Professor Hans Heesterbeek
mailto: proceedingsb@royalsociety.org

Associate Editor
Board Member: 1

Comments to Author:

This is a generally well-written and clear manuscript that describes a well-thought out and generally well-analyzed study. The reviewers have some good suggestions for improvements particularly to the treatment of side bias and for improving the clarity of model descriptions. A more major issue, in my mind, is the point that one reviewer made about previous published work that is highly relevant having been missed and as a result, the novelty of these findings being oversold. This is something that will take some careful thought and reading to remedy, but in the end I feel confident that the authors can highlight the truly novel contributions of this paper while also putting it in the context of all of the relevant literature.

Reviewer(s)' Comments to Author:

Referee: 1

Comments to the Author(s)

In the manuscript entitled „The importance of population heterogeneities in detecting social learning as the foundation of animal cultural transmission“ the authors report about a study in which the degree of female mate-choice copying was linked to the copying female’s personality type. The authors found that only distinct personality types copy the mate choice of other females in their experimental design, whereas other personality types were not responsive to this specific type of social learning. An interesting aspect that the authors stress is that behavioural differences among personality types can potentially blur the results in behavioural studies, which can lead to misinterpretations if this ubiquitous heterogeneity is ignored.

The manuscript is well-written and was easy to follow.

My advice for improvement of the manuscript is listed in a point-by-point fashion below:

The statement of novelty of the study seems very exaggerated given that a bunch of studies exist that investigated the link between socially-dependent mate choice (also in particular mate choice copying) and personality before. Those studies are not even cited although they are highly relevant. For example:

Gomes-Silva et al. 2017 “ Does individual variation in male mate choice copying reflect differences in social responsiveness?”

White et al. 2017 “Socialbility affects the intensity of mate-choice copying in female guppies, *Poecilia reticulata*”

Bierbach et al. 2015 “Personality affects mate choice: bolder males show stronger audience effects under high competition...etc.

Since the authors are competent in the study of animal behaviour and, especially, animal personality, I find it rather surprising that they seem to have overseen some of the most relevant studies in their context, which is a major problem that needs to be addressed. I strongly recommend the authors to include the existing literature and evaluate again what aspects of novelty their study provides compared to the already existing ones.

I recommend to replace “mate-copying” by “mate-choice copying”. Even though both terms appear in the scientific literature, the second one is, in my opinion, more correct and less misleading especially for readers unfamiliar with this field of research.

l. 55-56: I understand that the authors want to set the existence of these behavioural heterogeneities in a bigger context, however I felt very distracted by the detailed example about breeding habitat choice, which is not the topic of this study. I am sure it is possible to find a scenario that is closer to the actual research topic.

l. 118: what does “kept separately” mean? Individually (isolated) or separated by sex but still in groups? Please specify.

l. 212: Why was boldness not measured as a continuous variable? This would come with less information loss.

l. 217: That is a very high proportion of females. Do the authors have any explanation for this? Lightening, room etc.

l. 218: Is the dependent variable normally-distributed? Please show the statistical values.

l. 225: How many variables were included in the starting model? Although the authors used a rather big sample size, too many variables can cause overfitting of the model. The rule of thumb here would be to include variables of the number square root of sample size.

l. 291: Was that really unexpected? If I am not wrong the formula (MCT2-MCT1) that was used to calculate SLS is inherently larger for individuals that do not show much of a preference during the first test. This should be the case for females with strong side-bias. It seems to me that a female, which did have a strong preference in the first mate choice test (before copying), is unlikely to achieve a high SLS given this calculation. Please clarify.

Referee: 2

Comments to the Author(s)

Review of MS PSPB-2021-1567

This is a very interesting study, that shows how personality can affect the propensity of females to mate copy.

This is important for two reasons: 1) bold-shy dichotomy has mostly been discussed in foraging contexts, here it is shown to be of importance in the propensity of social learning in mate selection context.

2) Theory on the effect of mate choice copying has assumed that this would produce strong positive frequency dependent selection, which would wipe out variation in mating decisions, and there was until now no good reason to assume mechanisms to maintain variation in mate preferences in the population. The pattern shown in this paper adds a source of variation in mate preferences, that has not previously been discussed: copying from shy individuals could yield on a population level a source of positive frequency dependent selection, but the reluctance of bold females to copy prevents it sweep the population, maintaining an independent source of mate preferences.

The paper is overall well written, and I have no structural comments about the study design, or analysis. I do have a couple comments that I think would improve the paper.

1) In the introduction, the authors use a lot of space to draw an analogy to nest-site selection. I find this distracting to the topic at hand, and I would suggest the authors focus on mate selection instead, as I have described above. As it stands now, the authors don't make this point either in the introduction or discussion in the context of mate choice, and I feel this is a lost opportunity on their part.

2) In the results section, a large role is assigned to side bias scores. The authors need to justify this better.

First of all, does the side bias score correlate in any way with the personality score? I would like the authors to look at one or both of the following options: did side bias correlate with the time a female used to reach the last chamber in the boldness test? I am aware that the authors scored the females in a binary fashion as shy or bold, but perhaps they maintained a record of the time score of each female. If not, please check if shy individuals more likely to have a high side bias score?

Secondly, I would like to see some reflection on what a side bias may indicate, more than that an individual spent a lot of time on one side of the tank. In particular I would like the authors to consider that a high side bias score may indicate a lack of choice for a mate, a weak preference. If a female does not have a strong preference for a male, she might play it safe and stay away from males, avoiding predation risks, and/or avoiding male harassment. More importantly for this study, a female with a weak preference may be more likely to copy than a female that showed a strong preference. This point is important, because it relates to the underlying mechanism for why and when individuals use social learning. Shy individuals may not be 'better social learners', but those with a strong side bias are more likely to copy because they are less biased by prior preferences (prior to seeing the demonstrator choices). This also underscores why I would really like to see the author disentangle the effect of the side bias from the shy-bold scores better, because from this manuscript, I cannot be sure that there is a difference.

3) Lastly, I would like to see figure 3 improved. As it stands, there is no figure in the manuscript that visually shows the strong effect the authors found in the difference between shy-bold individuals and their copying score. I am not blown away by figure 3A, if I were ever to present this mechanism in a lecture or other type of presentation, for instance.

Minor comments:

Ln 62: "Similarly, the addition a small proportion" should read: "Similarly, the addition of a small proportion"

Ln 119: "Fish returned to the stock tanks after experiment." Should read: "Fish returned to the stock tanks after the experiment."

Ln 145: "... prefer the stimulus male close to which she..." should read: "... prefer the stimulus male closest to which she..."

Ln 212: please indicate the N for bold and shy females

Ln 218: please indicate the final N for each experiment after exclusion of the 90% side biased females.

Author's Response to Decision Letter for (RSPB-2021-1567.R0)

See Appendix A.

RSPB-2022-0431.R0

Review form: Reviewer 1

Recommendation

Accept with minor revision (please list in comments)

Scientific importance: Is the manuscript an original and important contribution to its field?

Acceptable

General interest: Is the paper of sufficient general interest?

Good

Quality of the paper: Is the overall quality of the paper suitable?

Acceptable

Is the length of the paper justified?

Yes

Should the paper be seen by a specialist statistical reviewer?

No

Do you have any concerns about statistical analyses in this paper? If so, please specify them explicitly in your report.

No

It is a condition of publication that authors make their supporting data, code and materials available - either as supplementary material or hosted in an external repository. Please rate, if applicable, the supporting data on the following criteria.

Is it accessible?

Yes

Is it clear?

Yes

Is it adequate?

No

Do you have any ethical concerns with this paper?

No

Comments to the Author

All in all, the authors did a good job in improving their manuscript. Their answers and improvements are largely adequate. I only have a few comments that I would like to see addressed before I can recommend publication:

The new paragraph in the introduction (l. 54-63 track changes version) is a bit unsatisfactory in its phrasing in my opinion. Firstly, the benefits and disadvantages of social and private information use are contrasted. Here it seems like social information use has many more benefits than private information use except for if too many individuals of the population rely on it. Social information is also prone to be incorrect and its reliability is dependent on the sending individual (e.g., its experience and honesty). Furthermore, relying on the same information as others should come with increased competition over the respective resource, at least when social information leads to a copying behaviour.

Also the conclusive sentence in line 61 does not really fit here. I would recommend a sentence that sums up the outcome for the individual (rather than for the group). For example, the previously explained frequency-dependence drives maintenance of both behavioral types in the population.

This maintenance of variance can of course be beneficial for the population but how the authors put it, it sounds more like a group selection kind of sentence.

After this paragraph, the authors switch to a sentence that seems very disconnected from the previous paragraph (line 64ff). It states that information use is dependent on intrinsic characteristics (only, at least this is how it sounds to me). This sentence should be less strict because otherwise the whole previous paragraph does not make much sense.

All in all I understand in which direction the authors wanted to improve the introduction here but I think the new paragraph should be elaborated on to better fit in the context and to be more sound on its own.

Line 332: I would recommend to use simply "they didn't find". Otherwise it accidentally sounds like finding no correlation is due to a failed experimental design.

l. 400. Start this sentence with something like "personality-dependent differences in information use", instead of "this" because it is very misleading since it points to the previous sentence, which doesn't explain why not all individuals copy.

l. 404 This statement is certainly too strong. Rather use something like "however, usually we see a broad variety of phenotypes etc."

Review form: Reviewer 2

Recommendation

Accept with minor revision (please list in comments)

Scientific importance: Is the manuscript an original and important contribution to its field?

Good

General interest: Is the paper of sufficient general interest?

Good

Quality of the paper: Is the overall quality of the paper suitable?

Good

Is the length of the paper justified?

Yes

Should the paper be seen by a specialist statistical reviewer?

No

Do you have any concerns about statistical analyses in this paper? If so, please specify them explicitly in your report.

No

It is a condition of publication that authors make their supporting data, code and materials available - either as supplementary material or hosted in an external repository. Please rate, if applicable, the supporting data on the following criteria.

Is it accessible?

Yes

Is it clear?

Yes

Is it adequate?

Yes

Do you have any ethical concerns with this paper?

No

Comments to the Author

This paper has improved very much from the first submission, and the authors have dealt with all issues I have raised in the first review.

Unfortunately, I have one additional comment. Although this manuscript is about the change of preferences, I feel that the actual preferences of the females should be mentioned. Do females in the controls actually prefer larger males? Do bold or shy females prefer the larger males when they are not copying? I think a sentence or two in the results section would suffice.

I understand fully this is not the focus of this paper, but I feel that it's important to state that this study confirms the earlier findings that, in general, females prefer larger males. I've already peeked quickly at the data, it seems like females in general prefer the larger males, so I am not doubting that this is a problem with this study. I just would like to see it stated somewhere in the manuscript.

I found the following three small typos:

In 49: ... in a form of density a dependence process; should read: in a form of a density dependent process.

In 211: We removed 0.5 to the absolute... should read: We removed 0.5 from the absolute

In 262: ... because removing it increase... should read: because removing it increased....

Decision letter (RSPB-2022-0431.R0)

25-Apr-2022

Dear Dr Nöbel:

Your manuscript has now been peer reviewed and the reviews have been assessed by an Associate Editor. The reviewers' comments (not including confidential comments to the Editor) and the comments from the Associate Editor are included at the end of this email for your reference. As you will see, the reviewers and the Associate Editor have raised some issues with your manuscript and we would like to invite you to revise your manuscript to address them.

When submitting your revision please upload a file under "Response to Referees" - in the "File Upload" section. This should document, point by point, how you have responded to the

reviewers' and Editors' comments, and the adjustments you have made to the manuscript. We also require a copy of the revised manuscript showing track changes to be uploaded.

Research ethics:

Use of animals and field studies:

It is a condition of publication that data supporting your paper are made available either in the electronic supplementary material. Authors must complete the 'data accessibility' section in the submission system. This should list the database and accession number for all data from the article that has been made publicly available, for instance:

NB. From April 1 2013, peer reviewed articles based on research funded wholly or partly by RCUK must include, if applicable, a statement on how the underlying research materials – such as data, samples or models – can be accessed.

[http://datadryad.org/submit?journalID=RSPB&manu=\(Document not available\)](http://datadryad.org/submit?journalID=RSPB&manu=(Document not available)) which will take you to your unique entry in the Dryad repository. If you have already submitted your data to dryad you can make any necessary revisions to your dataset by following the above link.

Please include the Dryad DOI in the Data Accessibility section and reference in the paper's bibliography.

Please see our Data Sharing Policies (<https://royalsociety.org/journals/authors/author-guidelines/>).

Online supplementary material will also carry the title and description provided during submission, so please ensure these are accurate and informative. Note that the Royal Society will

not edit or typeset supplementary material and it will be hosted as provided. Please ensure that the supplementary material includes the paper details (authors, title, journal name, article DOI). Your article DOI will be 10.1098/rspb.[paper ID in form xxxx.xxxx e.g. 10.1098/rspb.2016.0049].

Please submit a copy of your revised paper within three weeks. If we do not hear from you within this time your manuscript will be rejected. If you are unable to meet this deadline please let us know as soon as possible, as we may be able to grant a short extension.

Best wishes,
Professor Hans Heesterbeek
mailto:proceedingsb@royalsociety.org

Associate Editor Board Member
Comments to Author:

Thank you for your careful revisions of this manuscript. With the exception of some revisions to the introduction and results sections suggested by the reviewers and a few small wording issues to be corrected, the paper appears to be in good shape. Please consider making the minor changes suggested by the reviewers and if you choose not to, detail the reasons why not in your rebuttal.

Reviewer(s)' Comments to Author:

Referee: 1

Comments to the Author(s).

All in all, the authors did a good job in improving their manuscript. Their answers and improvements are largely adequate. I only have a few comments that I would like to see addressed before I can recommend publication:

The new paragraph in the introduction (l. 54-63 track changes version) is a bit unsatisfactory in its phrasing in my opinion. Firstly, the benefits and disadvantages of social and private information use are contrasted. Here it seems like social information use has many more benefits than private information use except for if too many individuals of the population rely on it. Social information is also prone to be incorrect and its reliability is dependent on the sending individual (e.g., its experience and honesty). Furthermore, relying on the same information as others should come with increased competition over the respective resource, at least when social information leads to a copying behaviour.

Also the conclusive sentence in line 61 does not really fit here. I would recommend a sentence that sums up the outcome for the individual (rather than for the group). For example, the previously explained frequency-dependence drives maintenance of both behavioral types in the population.

This maintenance of variance can of course be beneficial for the population but how the authors put it, it sounds more like a group selection kind of sentence.

After this paragraph, the authors switch to a sentence that seems very disconnected from the previous paragraph (line 64ff). It states that information use is dependent on intrinsic characteristics (only, at least this is how it sounds to me). This sentence should be less strict because otherwise the whole previous paragraph does not make much sense.

All in all I understand in which direction the authors wanted to improve the introduction here but I think the new paragraph should be elaborated on to better fit in the context and to be more sound on its own.

Line 332: I would recommend to use simply "they didn't find". Otherwise it accidentally sounds like finding no correlation is due to a failed experimental design.

l. 400. Start this sentence with something like “personality-dependent differences in information use”, instead of “this” because it is very misleading since it points to the previous sentence, which doesn’t explain why not all individuals copy.

l. 404 This statement is certainly too strong. Rather use something like “however, usually we see a broad variety of phenotypes etc.”

Referee: 2

Comments to the Author(s).

This paper has improved very much from the first submission, and the authors have dealt with all issues I have raised in the first review.

Unfortunately, I have one additional comment. Although this manuscript is about the change of preferences, I feel that the actual preferences of the females should be mentioned. Do females in the controls actually prefer larger males? Do bold or shy females prefer the larger males when they are not copying? I think a sentence or two in the results section would suffice.

I understand fully this is not the focus of this paper, but I feel that it's important to state that this study confirms the earlier findings that, in general, females prefer larger males. I've already peeked quickly at the data, it seems like females in general prefer the larger males, so I am not doubting that this is a problem with this study. I just would like to see it stated somewhere in the manuscript.

I found the following three small typos:

In 49: ... in a form of density a dependence process; should read: in a form of a density dependent process.

In 211: We removed 0.5 to the absolute... should read: We removed 0.5 from the absolute

In 262: ... because removing it increase... should read: because removing it increased....

Author's Response to Decision Letter for (RSPB-2022-0431.R0)

See Appendix B.

Decision letter (RSPB-2022-0431.R1)

17-May-2022

Dear Dr Nöbel

I am pleased to inform you that your manuscript entitled "The importance of population heterogeneities in detecting social learning as the foundation of animal cultural transmission" has been accepted for publication in Proceedings B.

Data Accessibility section

Open Access

Paper charges

Sincerely,

Professor Hans Heesterbeek

Associate Editor:

Board Member

Comments to Author:

(There are no comments.)

Appendix A

INSTITUTE for
ADVANCED
STUDY in
TOULOUSE

Toulouse March 4th, 2022

Dear Professor Heesterbeek,

We hereby submit a thoroughly revised version of our manuscript “The importance of population heterogeneities in detecting social learning as the foundation of animal cultural transmission” for publication in *Proceedings of the Royal Society B*. The manuscript has been submitted before (#RSPB-2021-1567) and we appreciate that you suggested a new submission of the revised manuscript. The manuscript carefully accounts for all your comments and those of the referees, in a way that we describe below, with referees' comments in black and our replies in blue.

As a reminder, many studies have predicted and documented high levels of within-population behavioural variation which can have drastic demographic consequences, thus changing the evolutionary fate of populations. A major source of within-population heterogeneity is personality which has been documented in many organisms and is often measured in terms of boldness. Nonetheless, the impact of population heterogeneity is often neglected in social learning (i.e., learning from others) studies that constitutes the most basic process of cultural transmission. Here, we perform in female mosquitofish (*Gambusia holbrooki*) a social learning experiment in the context of mate choice, called mate copying and for which there is strong evidence that it can lead to the emergence of persistent cultural traditions of preferring a given male phenotype. When accounting for the natural tendency of mosquitofish females to prefer larger males but ignoring differences in personality within our sample, we detected no evidence for mate copying. However, when accounting for the bold-shy dichotomy, we found that bold females did not show any evidence for mate copying, while shy females showed significant amounts of mate copying. This illustrates how the presence of variation in personality within a population can hamper our capacity to detect mate copying. We conclude that, mate copying may be far more widespread than we thought because the fact that many studies ignored the presence of heterogeneities within the study population might have hidden the existence of mate copying leading to false negative tests.

We hope that you will find this new version satisfactory. We appreciate the reviewer comments on the previous version that greatly helped improving the manuscript and rethinking some sections.

We are looking forward to your opinion about whether you would consider this revised manuscript in *Proceedings of the Royal Society B*.

Sincerely,
Sabine Nöbel

Associate Editor

Board Member: 1

Comments to Author:

This is a generally well-written and clear manuscript that describes a well-thought out and generally well-analyzed study. The reviewers have some good suggestions for improvements particularly to the treatment of side bias and for improving the clarity of model descriptions. A more major issue, in my mind, is the point that one reviewer made about previous published work that is highly relevant having been missed and as a result, the novelty of these findings being oversold. This is something that will take some careful thought and reading to remedy, but in the end I feel confident that the authors can highlight the truly novel contributions of this paper while also putting it in the context of all of the relevant literature.

We appreciate your suggestions and apologize for having overlooked some papers that we now cite in the introduction and the discussion. We also toned down the novelty statement in the entire manuscript. However, we still think that our study contributes some new aspects to the field.

Reviewer(s)' Comments to Author:

Referee: 1

Comments to the Author(s)

In the manuscript entitled „The importance of population heterogeneities in detecting social learning as the foundation of animal cultural transmission” the authors report about a study in which the degree of female mate-choice copying was linked to the copying female’s personality type. The authors found that only distinct personality types copy the mate choice of other females in their experimental design, whereas other personality types were not responsive to this specific type of social learning. An interesting aspect that the authors stress is that behavioural differences among personality types can potentially blur the results in behavioural studies, which can lead to misinterpretations if this ubiquitous heterogeneity is ignored.

The manuscript is well-written and was easy to follow.

My advice for improvement of the manuscript is listed in a point-by-point fashion below:

The statement of novelty of the study seems very exaggerated given that a bunch of studies exist that investigated the link between socially-dependent mate choice (also in particular mate choice copying) and personality before. Those studies are not even cited although they are highly relevant.

For example:

Gomes-Silva et al. 2017 “ Does individual variation in male mate choice copying reflect differences in social responsiveness?”

White et al. 2017 “Socialbility affects the intensity of mate-choice copying in female guppies, *Poecilia reticulata*”

Bierbach et al. 2015 “Personality affects mate choice: bolder males show stronger audience effects under high competition...etc.

Since the authors are competent in the study of animal behaviour and, especially, animal personality, I find it rather surprising that they seem to have overseen some of the most relevant studies in their context, which is a major problem that needs to be addressed. I strongly recommend

the authors to include the existing literature and evaluate again what aspects of novelty their study provides compared to the already existing ones.

Thanks for reminding us of these studies. We cite them now in the introduction lines 77/93 and the discussion (lines 329f). The novelty statement is diminished throughout the entire manuscript.

I recommend to replace “mate-copying” by “mate-choice copying”. Even though both terms appear in the scientific literature, the second one is, in my opinion, more correct and less misleading especially for readers unfamiliar with this field of research.

We don't really understand why the reviewer thinks that the term 'mate copying' is misleading (for a discussion see Wagner & Danchin 2010, Danchin et al. 2021). As both terms are commonly used, we would prefer to keep mate copying.

I. 55-56: I understand that the authors want to set the existence of these behavioural heterogeneities in a bigger context, however I felt very distracted by the detailed example about breeding habitat choice, which is not the topic of this study. I am sure it is possible to find a scenario that is closer to the actual research topic.

We removed the example of breeding habitat choice and replaced it with an example of mate choice as suggested. The paragraph reads now (lines 53f):

“In the context of mate choice, if populations had only one type of strategy, for instance, “choose your mate based on private information“, this would imply high sampling costs before choosing a mate. Beside high energetic costs, this sampling strategy includes predation risk, the risk of harassment, as well as a high risk of failure due to sampling errors (reviewed in [5]). Using social information to assess the quality of mates can increase accuracy and reduce the costs of gathering private information on a number of potential mates [6]. However, if all individuals in a population choose their mates based on publicly available social information the risk is high that this information is quickly outdated in rapidly changing environments. Thus, a natural source of heterogeneity leading groups to encompass individual and social learners should perform better in reacting to environmental changes.”

I. 118: what does “kept separately” mean? Individually (isolated) or separated by sex but still in groups? Please specify.

The fish were sexed and then kept in same sex groups (see line 118).

I. 212: Why was boldness not measured as a continuous variable? This would come with less information loss.

We used the shy-bold dichotomy to have more power in the analysis.

I. 217: That is a very high proportion of females. Do the authors have any explanation for this? Lightening, room etc.

We don't think that excluding 16% of the females is a high number and the 90% threshold is common in the literature. Females are not always receptive and interested in males, especially as *Gambusia* males don't perform special courtship to attract females and harassment rate is high. We

couldn't find any systematic pattern in the side bias which could be explained by room/light conditions or the behaviour of the experimenter.

I. 218: Is the dependent variable normally-distributed? Please show the statistical values.

The dependent variable is normally distributed, and we provide the p-value now in line 238.

I. 225: How many variables were included in the starting model? Although the authors used a rather big sample size, too many variables can cause overfitting of the model. The rule of thumb here would be to include variables of the number square root of sample size.

With a sample size of 285 females in all three treatments we could have started with 16 variables due to the rule of thumb. However, we started with 5 variables (treatment (t), personality (p), side-bias, size (s) and size²) and 4 interaction terms (t:p:s, t:p, t:s and p:s) as described in the method section (lines 237-242).

I. 291: Was that really unexpected? If I am not wrong the formula (MCT2-MCT1) that was used to calculate SLS is inherently larger for individuals that do not show much of a preference during the first test. This should be the case for females with strong side-bias. It seems to me that a female, which did have a strong preference in the first mate choice test (before copying), is unlikely to achieve a high SLS given this calculation. Please clarify.

We removed the "unexpected" and apologize. The reviewer is right here.

Referee: 2

Comments to the Author(s)

Review of MS PSPB-2021-1567

This is a very interesting study, that shows how personality can affect the propensity of females to mate copy.

This is important for two reasons: 1) bold-shy dichotomy has mostly been discussed in foraging contexts, here it is shown to be of importance in the propensity of social learning in mate selection context.

2) Theory on the effect of mate choice copying has assumed that this would produce strong positive frequency dependent selection, which would wipe out variation in mating decisions, and there was until now no good reason to assume mechanisms to maintain variation in mate preferences in the population. The pattern shown in this paper adds a source of variation in mate preferences, that has not previously been discussed: copying from shy individuals could yield on a population level a source of positive frequency dependent selection, but the reluctance of bold females to copy prevents it sweep the population, maintaining an independent source of mate preferences.

Many thanks for these very useful comments. We used them to improve our discussion (lines 400f).

The paper is overall well written, and I have no structural comments about the study design, or analysis. I do have a couple comments that I think would improve the paper.

1) In the introduction, the authors use a lot of space to draw an analogy to nest-site selection. I find

this distracting to the topic at hand, and I would suggest the authors focus on mate selection instead, as I have described above. As it stands now, the authors don't make this point either in the introduction or discussion in the context of mate choice, and I feel this is a lost opportunity on their part.

We removed the example of breeding habitat choice and replaced it with an example of mate choice as suggested. The paragraph reads now (lines 53f):

“In the context of mate choice, if populations had only one type of strategy, for instance, “choose your mate based on private information“, this would imply high sampling costs before choosing a mate. Beside high energetic costs, this sampling strategy includes predation risk, the risk of harassment, as well as a high risk of failure due to sampling errors (reviewed in [5]). Using social information to assess the quality of mates can increase accuracy and reduce the costs of gathering private information on a number of potential mates [6]. However, if all individuals in a population choose their mates based on publicly available social information the risk is high that this information is quickly outdated in rapidly changing environments. Thus, a natural source of heterogeneity leading groups to encompass individual and social learners should perform better in reacting to environmental changes.”

2) In the results section, a large role is assigned to side bias scores. The authors need to justify this better.

First of all, does the side bias score correlate in any way with the personality score? I would like the authors to look at one or both of the following options: did side bias correlate with the time a female used to reach the last chamber in the boldness test? I am aware that the authors scored the females in a binary fashion as shy or bold, but perhaps they maintained a record of the time score of each female. If not, please check if shy individuals more likely to have a high side bias score?

We tried to explore the side-bias score in paragraph lines 357f. In our dataset, the side-bias of shy and bold females did not differ statistically (Mann-Whitney U test: $W = 2200$, $P = 0.886$). Thus, the side-bias score must measure something different than the shy-bold dichotomy, and both affect mate copying.

Secondly, I would like to see some reflection on what a side bias may indicate, more than that an individual spent a lot of time on one side of the tank. In particular, I would like the authors to consider that a high side bias score may indicate a lack of choice for a mate, a weak preference. If a female does not have a strong preference for a male, she might play it safe and stay away from males, avoiding predation risks, and/or avoiding male harassment. More importantly for this study, a female with a weak preference may be more likely to copy than a female that showed a strong preference. This point is important, because it relates to the underlying mechanism for why and when individuals use social learning. Shy individuals may not be 'better social learners', but those with a strong side bias are more likely to copy because they are less biased by prior preferences (prior to seeing the demonstrator choices). This also underscores why I would really like to see the author disentangle the effect of the side bias from the shy-bold scores better, because from this manuscript, I cannot be sure that there is a difference.

Many thanks for these very useful suggestions. We included part of them now together with the comments above in the discussion (lines 370f, lines 400f). However, we don't think that the mentioned "playing safe" argument holds true in our situation as we calculated the side-bias based on the time in a mate-choice zone, thus, next to a male. Thus, a high score means she spent a lot of time next to a male which doesn't prevent harassment.

3) Lastly, I would like to see figure 3 improved. As it stands, there is no figure in the manuscript that visually shows the strong effect the authors found in the difference between shy-bold individuals and their copying score. I am not blown away by figure 3A, if I were ever to present this mechanism in a lecture or other type of presentation, for instance.

We have thought for a long time about how to make Figure 3 clearer. Showing only the social learning index separated by personality seems to us a too big simplification, since it completely neglects the different male size-ratios and the side-bias. In fact, the current figure is also only a compromise, as it ignores the side-bias aspect. We have integrated this in Figure 4.

Minor comments:

Ln 62: "Similarly, the addition a small proportion" should read: "Similarly, the addition of a small proportion"

This sentence has been removed completely as we removed the entire paragraph with the breeding habitat example.

Ln 119: "Fish returned to the stock tanks after experiment." Should read: "Fish returned to the stock tanks after the experiment."

We changed the sentence accordingly to your suggestion.

Ln 145: "... prefer the stimulus male close to which she..." should read: "... prefer the stimulus male closest to which she..."

Thanks for spotting this mistake. We changed it accordingly to your suggestion.

Ln 212: please indicate the N for bold and shy females

We added this information and a reference to our supplementary material (line 224/225).

Ln 218: please indicate the final N for each experiment after exclusion of the 90% side biased females.

The numbers were already correct, but we clarified our phrasing (lines 229f).

Appendix B

Reviewer(s)' Comments to Author:

Referee: 1

Comments to the Author(s).

All in all, the authors did a good job in improving their manuscript. Their answers and improvements are largely adequate. I only have a few comments that I would like to see addressed before I can recommend publication:

The new paragraph in the introduction (l. 54-63 track changes version) is a bit unsatisfactory in its phrasing in my opinion. Firstly, the benefits and disadvantages of social and private information use are contrasted. Here it seems like social information use has many more benefits than private information use except for if too many individuals of the population rely on it. Social information is also prone to be incorrect and its reliability is dependent on the sending individual (e.g., its experience and honesty). Furthermore, relying on the same information as others should come with increased competition over the respective resource, at least when social information leads to a copying behaviour.

Also the conclusive sentence in line 61 does not really fit here. I would recommend a sentence that sums up the outcome for the individual (rather than for the group). For example, the previously explained frequency-dependence drives maintenance of both behavioral types in the population.

This maintenance of variance can of course be beneficial for the population but how the authors put it, it sounds more like a group selection kind of sentence.

After this paragraph, the authors switch to a sentence that seems very disconnected from the previous paragraph (line 64ff). It states that information use is dependent on intrinsic characteristics (only, at least this is how it sounds to me). This sentence should be less strict because otherwise the whole previous paragraph does not make much sense.

All in all I understand in which direction the authors wanted to improve the introduction here but I think the new paragraph should be elaborated on to better fit in the context and to be more sound on its own.

We really appreciated your comments and thank you for reviewing our manuscript again. We changed the text according to your suggestions into:

“Such heterogeneities can have drastic consequences in terms of population dynamics, and thus can change the adaptive and evolutionary fate of populations. In the context of mate choice, if populations had only one type of strategy, for instance, “choose your mate based on private information“, this would imply high sampling costs before choosing a mate. Beside high energetic costs, this sampling strategy includes predation risk, the risk of harassment, as well as a high risk of failure due to sampling errors (reviewed in [5]). Using social information to assess the quality of mates can increase accuracy and reduce the costs of gathering private information on a number of potential mates [6]. However, to receive valid social information

the observing individual need to rely on honest signalling from the sender and slow environmental changes as otherwise the information is quickly outdated. Furthermore, it may create competition over resources if many individuals rely on the same information. Since both the use of private and social information has advantages and disadvantages for the individual as well as for the group, which depend, among other things, on how many individuals in a population use each strategy, both strategies are expected in natural populations.” (lines 52ff) and hope it is satisfying now.

Line 332: I would recommend to use simply “they didn’t find”. Otherwise it accidentally sounds like finding no correlation is due to a failed experimental design.

We corrected the sentence as suggested (line 338).

I. 400. Start this sentence with something like “personality-dependent differences in information use”, instead of “this” because it is very misleading since it points to the previous sentence, which doesn’t explain why not all individuals copy.

We changed the sentence into “If personality-dependent differences in information use proved to be valid, this would further imply that the incidence of mate copying might be much broader than usually thought.” (line 405).

I. 404 This statement is certainly too strong. Rather use something like “however, usually we see a broad variety of phenotypes etc.”

We changed the sentence according to your suggestions to “However, usually we see a broad variety of phenotypes in nature, although there has been no explanation for a mechanism that maintains variation in mating preferences.” (lines 410/411).

Referee: 2

Comments to the Author(s).

This paper has improved very much from the first submission, and the authors have dealt with all issues I have raised in the first review.

We really appreciated your comments and thank you for reviewing our manuscript again.

Unfortunately, I have one additional comment. Although this manuscript is about the change of preferences, I feel that the actual preferences of the females should be mentioned. Do females in the controls actually prefer larger males? Do bold or shy females prefer the larger males when they are not copying? I think a sentence or two in the results section would suffice.

I understand fully this is not the focus of this paper, but I feel that it's important to state

that this study confirms the earlier findings that, in general, females prefer larger males. I've already peeked quickly at the data, it seems like females in general prefer the larger males, so I am not doubting that this is a problem with this study. I just would like to see it stated somewhere in the manuscript.

We analyzed the time spent in front of the large and small male in each mate-choice test as a measure for females' preference. We used only cases in which males differed in size and the mate-copying treatment plus control 1 because in control 2 only female stimuli were used. Indeed, females in the mate-copying treatment and control 1 preferred larger males before receiving social information and only shy females lost this preference after seeing the smaller male with another female.

We added a sentence to the results section: "Without any social information, females significantly preferred larger males in the first mate-choice test of the mate-copying and control 1 treatment (i.e., prior to receiving any social information; Wilcoxon test: $N = 184$, $V = 12936$, $P < 0.0001$; for more information, please see table 4 in the supplementary material).", lines 261ff) and provide further information in the supplementary material.

I found the following three small typos:

In 49: ... in a form of density a dependence process; should read: in a form of a density dependent process.

In 211: We removed 0.5 to the absolute... should read: We removed 0.5 from the absolute

In 262: ... because removing it increase... should read: because removing it increased....

We corrected all three typos and apologize (lines 49, 213, 268).